# OFFLINE Q-LEARNING ON DIVERSE MULTI-TASK DATA BOTH SCALES AND GENERALIZES

**Aviral Kumar**[1,2]     **Rishabh Agarwal**[1]
**Xinyang Geng**[2]     **George Tucker**[*,1]     **Sergey Levine**[*,1,2]
[1] Google Research, Brain Team     [2] UC Berkeley
{aviralk, young.geng, svlevine}@eecs.berkeley.edu, {rishabhagarwal, gjt}@google.com

## ABSTRACT

The potential of offline reinforcement learning (RL) is that high-capacity models trained on large, heterogeneous datasets can lead to agents that generalize broadly, analogously to similar advances in vision and NLP. However, recent works argue that offline RL methods encounter unique challenges to scaling up model capacity. Drawing on the learnings from these works, we re-examine previous design choices and find that with appropriate choices: *ResNets, cross-entropy based distributional backups, and feature normalization*, offline Q-learning algorithms exhibit strong performance that scales with model capacity. Using multi-task Atari as a test-bed for scaling and generalization, we train a single policy on 40 games with near-human performance using up-to 80 million parameter networks, finding that model performance scales favorably with capacity. In contrast to prior work, we extrapolate beyond dataset performance even when trained entirely on a large (400M transitions) but highly suboptimal dataset (51% human-level performance). Compared to return-conditioned supervised approaches, offline Q-learning scales similarly with model capacity and has better performance, especially when the dataset is suboptimal. Finally, we show that offline Q-learning with a diverse dataset is sufficient to learn powerful representations that facilitate rapid transfer to novel games and fast online learning on new variations of a training game, improving over existing state-of-the-art representation learning approaches.

## 1 INTRODUCTION

High-capacity neural networks trained on large, diverse datasets have led to remarkable models that can solve numerous tasks, rapidly adapt to new tasks, and produce general-purpose representations in NLP and vision (Brown et al., 2020; He et al., 2021). The promise of offline RL is to leverage these advances to produce polices with broad generalization, emergent capabilities, and performance that exceeds the capabilities demonstrated in the training dataset. Thus far, the only offline RL approaches that demonstrate broadly generalizing policies and transferable representations are heavily-based on supervised learning (Reed et al., 2022; Lee et al., 2022). However, these approaches are likely to perform poorly when the dataset does not contain expert trajectories (Kumar et al., 2021b).

Offline Q-learning performs well across dataset compositions in a variety of simulated (Gulcehre et al., 2020; Fu et al., 2020) and real-world domains (Chebotar et al., 2021; Soares et al., 2021), however, these are largely centered around small-scale, single-task problems where broad generalization and learning general-purpose representations is not expected. *Scaling these methods up to high-capcity models on large, diverse datasets is the critical challenge.* Prior works hint at the difficulties: on small-scale, single-task deep RL benchmarks, scaling model capacity can lead to instabilities or degrade performance (Van Hasselt et al., 2018; Sinha et al., 2020; Ota et al., 2021) explaining why decade-old tiny 3-layer CNN architectures (Mnih et al., 2013) are still prevalent. Moreover, works that have scaled architectures to millions of parameters (Espeholt et al., 2018; Teh et al., 2017; Vinyals et al., 2019; Schrittwieser et al., 2021) typically focus on *online* learning and employ many sophisticated techniques to stabilize learning, such as supervised auxiliary losses, distillation, and pre-training. Thus, it is unclear whether offline Q-learning can be scaled to high-capacity models.

---

[*] Co-senior authors.

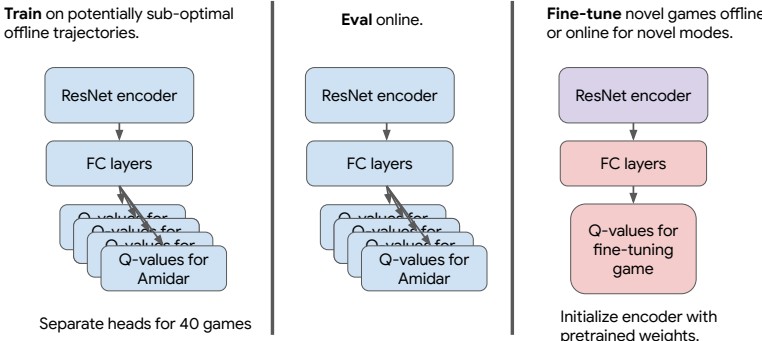

Figure 1: An overview of the training and evaluation setup. Models are trained offline with potentially sub-optimal data. We adapt CQL to the multi-task setup via a multi-headed architecture. The pre-trained visual encoder is reused in fine-tuning (the weights are either frozen or fine-tuned), whereas the downstream fully-connected layers are reinitialized and trained.

In this paper, we demonstrate that with careful design decisions, *offline Q-learning can scale* to high-capacity models trained on large, diverse datasets from many tasks, leading to policies that not only generalize broadly, but also learn representations that effectively transfer to new downstream tasks and exceed the performance in the training dataset. Crucially, we make three modifications motivated by prior work in deep learning and offline RL. First, we find that a modified ResNet architecture (He et al., 2016) substantially outperforms typical deep RL architectures and follows a power-law relationship between model capacity and performance, unlike common alternatives. Second, a discretized representation of the return distribution with a distributional cross-entropy loss (Bellemare et al., 2017) substantially improves performance compared to standard Q-learning, that utilizes mean squared error. Finally, feature normalization on the intermediate feature representations stabilizes training and prevents feature co-adaptation (Kumar et al., 2021a).

To systematically evaluate the impact of these changes on scaling and generalization, we train a single policy to play 40 Atari games (Bellemare et al., 2013; Agarwal et al., 2020), similarly to Lee et al. (2022), and evaluate performance when the training dataset contains expert trajectories *and* when the data is sub-optimal. This problem is especially challenging because of the diversity of games with their own unique dynamics, reward, visuals, and agent embodiments. Furthermore, the sub-optimal data setting requires the learning algorithm to "stitch together" useful segments of sub-optimal trajectories to perform well. To investigate generalization of learned representations, we evaluate offline fine-tuning to *never-before-seen* games and fast online adaptation on new *variants* of training games (Section 5.2). With our modifications,

- Offline Q-learning learns policies that attain more than 100% human-level performance on most of these games, about **2x** better than prior supervised learning (SL) approaches for learning from sub-optimal offline data (51% human-level performance).

- Akin to scaling laws in SL (Kaplan et al., 2020), offline Q-learning performance scales favorably with model capacity (Figure 6).

- Representations learned by offline Q-learning give rise to more than 80% better performance when fine-tuning on new games compared to representations from state-of-the-art return-conditioned supervised (Lee et al., 2022) and self-supervised methods (He et al., 2021; Oord et al., 2018).

By scaling Q-learning, we realize the promise of offline RL: learning policies that broadly generalize and exceed the capabilities demonstrated in the training dataset. We hope that this work encourages large-scale offline RL applications, especially in domains with large sub-optimal datasets.

## 2 RELATED WORK

Prior works have sought to train a single generalist policy to play multiple Atari games simultaneously from environment interactions, either using off-policy RL with online data collection (Espeholt et al., 2018; Hessel et al., 2019a; Song et al., 2019), or policy distillation (Teh et al., 2017; Rusu et al., 2015) from single-task policies. While our work also focuses on learning such a generalist multi-task

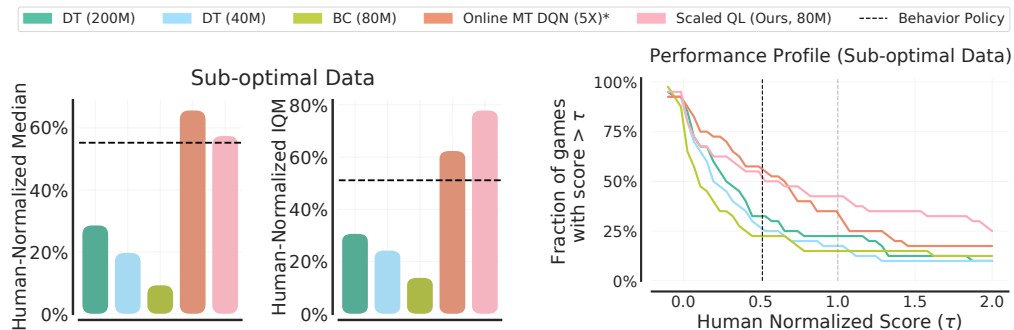

Figure 2: Offline multi-task performance on 40 games with sub-optimal data. **Left**. Scaled QL significantly outperforms the previous state-of-the-art method, DT, attaining about a **2.5x** performance improvement in normalized IQM score. To contextualize the absolute numbers, we include online multi-task Impala DQN (Espeholt et al., 2018) trained on 5x as much data. **Right**. Performance profiles (Agarwal et al., 2021) showing the distribution of normalized scores across all 40 training games (higher is better). Scaled QL stochastically dominates other offline RL algorithms and achieves superhuman performance in 40% of the games. "Behavior policy" corresponds to the score of the dataset trajectories. Online MT DQN (5X), taken directly from Lee et al. (2022), corresponds to running multi-task online RL for 5x more data with IMPALA (details in Appendix C.5).

policy, it investigates whether we can do so by scaling offline Q-learning on suboptimal offline data, analogous to how supervised learning can be scaled to large, diverse datasets. Furthermore, prior attempts to apply transfer learning using RL-learned policies in ALE (Rusu et al., 2015; Parisotto et al., 2015; Mittel & Sowmya Munukutla, 2019) are restricted to a dozen games that tend to be similar and generally require an "expert", instead of learning how to play all games concurrently.

Closely related to our work, recent work train Transformers (Vaswani et al., 2017) on purely offline data for learning such a generalist policy using supervised learning (SL) approaches, namely, behavioral cloning (Reed et al., 2022) or return-conditioned behavioral cloning (Lee et al., 2022). While these works focus on large datasets containing expert or near-human performance trajectories, our work focuses on the regime when we only have access to highly diverse but sub-optimal datasets. We find that these SL approaches perform poorly with such datasets, while offline Q-learning is able to substantially extrapolate beyond dataset performance (Figure 2). Even with near-optimal data, we observe that scaling up offline Q-learning outperforms SL approaches with 200 million parameters using as few as half the number of network parameters (Figure 6).

There has been a recent surge of offline RL algorithms that focus on mitigating distribution shift in single task settings (Fujimoto et al., 2018; Kumar et al., 2019; Liu et al., 2020; Wu et al., 2019; Fujimoto & Gu, 2021; Siegel et al., 2020; Peng et al., 2019; Nair et al., 2020; Liu et al., 2019; Swaminathan & Joachims, 2015; Nachum et al., 2019; Kumar et al., 2020; Kostrikov et al., 2021; Kidambi et al., 2020; Yu et al., 2020b; 2021). Complementary to such work, our work investigates scaling offline RL on the more diverse and challenging multi-task Atari setting with data from 40 different games (Agarwal et al., 2020; Lee et al., 2022). To do so, we use CQL (Kumar et al., 2020), due to its simplicity as well as its efficacy on offline RL datasets with high-dimensional observations.

## 3 PRELIMINARIES AND PROBLEM SETUP

We consider sequential-decision making problems (Sutton & Barto, 1998) where on each timestep, an agent observes a state $\mathbf{s}$, produces an action $\mathbf{a}$, and receives a reward $r$. The goal of a learning algorithm is to maximize the sum of discounted rewards. Our approach is based on conservative Q-learning (CQL) (Kumar et al., 2020), an offline Q-learning algorithm. CQL uses a sum of two loss functions to combat value overestimation on unseen actions: **(i)** standard TD-error that enforces Bellman consistency, and **(ii)** a regularizer that minimizes the Q-values for unseen actions at a given state, while maximizing the Q-value at the dataset action to counteract excessive underestimation. Denoting $Q_\theta(\mathbf{s}, \mathbf{a})$ as the learned Q-function, the training objective for CQL is given by:

$$\min_\theta \; \alpha \left( \mathbb{E}_{\mathbf{s}\sim\mathcal{D}} \left[ \log \left( \sum_{\mathbf{a}'} \exp(Q_\theta(\mathbf{s}, \mathbf{a}')) \right) \right] - \mathbb{E}_{\mathbf{s},\mathbf{a}\sim\mathcal{D}} \left[ Q_\theta(\mathbf{s}, \mathbf{a}) \right] \right) + \mathsf{TDError}(\theta; \mathcal{D}), \quad (1)$$

where $\alpha$ is the regularizer weight, which we fix to $\alpha = 0.05$ based on preliminary experiments unless noted otherwise. Kumar et al. (2020) utilized a distributional $\mathsf{TDError}(\theta; \mathcal{D})$ from C51 (Bellemare

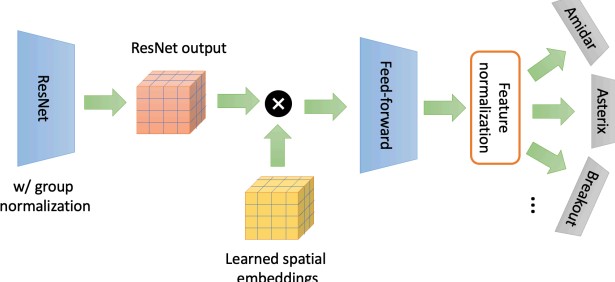

Figure 3: **An overview of the network architecture.** The key design decisions are: (1) the use of ResNet models with learned spatial embeddings and group normalization, (2) use of a distributional representation of return values and cross-entropy TD loss for training (i.e., C51 (Bellemare et al., 2017)), and (3) feature normalization to stablize training.

et al., 2017), whereas (Kumar et al., 2021a) showed that similar results could be attained with the standard mean-squared TD-error. Lee et al. (2022) use the distributional formulation of CQL and found that it underperforms alternatives and performance does not improve with model capacity. In general, there is no consensus on which formulation of TD-error must be utilized in Equation 1, and we will study this choice in our scaling experiments.

**Problem setup.** Our goal is to learn a single policy that is effective at multiple Atari games and can be fine-tuned to new games. For training, we utilize the set of 40 Atari games used by Lee et al. (2022), and for each game, we utilize the experience collected in the DQN-Replay dataset (Agarwal et al., 2020) as our offline dataset. We consider two different dataset compositions:

1. **Sub-optimal** dataset consisting of the initial 20% of the trajectories (10M transitions) from DQN-Replay for each game, containing 400 million transitions overall with average human-normalized interquartile-mean (IQM) (Agarwal et al., 2021) score of 51%. Since this dataset does not contain optimal trajectories, we do not expect methods that simply copy behaviors in this dataset to perform well. On the other hand, we would expect methods that can combine useful segments of sub-optimal trajectories to perform well.

2. **Near-optimal** dataset, used by Lee et al. (2022), consisting of all the experience (50M transitions) encountered during training of a DQN agent including human-level trajectories, containing 2 billion transitions with average human-normalized IQM score of 93.5%.

**Evaluation**. We evaluate our method in a variety of settings as we discuss in our experiments in Section 5. Due to excessive computational requirements of running huge models, we are only able to run our main experiments with one seed. Prior work (Lee et al., 2022) that also studied offline multi-game Atari evaluated models with only one seed. That said, to ensure that our evaluations are reliable, for reporting performance, we follow the recommendations by Agarwal et al. (2021). Specifically, we report interquartile mean (IQM) normalized scores, which is the average scores across middle 50% of the games, as well as performance profiles for qualitative summarization.

## 4    OUR APPROACH FOR SCALING OFFLINE RL

In this section, we describe the critical modifications required to make CQL effective in learning highly-expressive policies from large, heterogeneous datasets.

**Parameterization of Q-values and TD error.** In the single game setting, both mean-squared TD error and distributional TD error perform comparably online (Agarwal et al., 2021) and offline (Kumar et al., 2020; 2021a). In contrast, we observed, perhaps surprisingly, that mean-squared TD error does not scale well, and performs much worse than using a **categorical distributional representation of return values** (Bellemare et al., 2017) when we train on many Atari games. We hypothesize that this is because even with reward clipping, Q-values for different games often span different ranges, and training a single network with shared parameters to accurately predict all of them presents challenges pertaining to gradient interference along different games (Hessel et al., 2019b; Yu et al., 2020a). While prior works have proposed to use adaptive normalization schemes (Hessel et al., 2019b; Kurin et al., 2022), preliminary experiments with these approaches were not effective to close the gap.

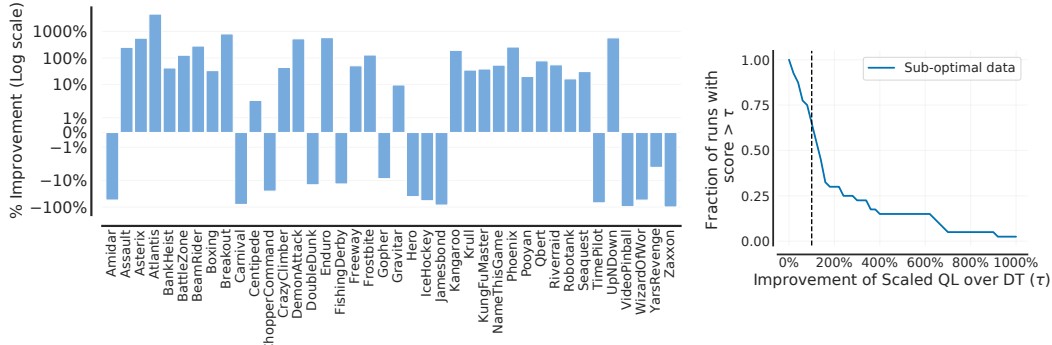

Figure 4: **Comparing Scaled QL to DT** on all training games on the sub-optimal dataset.

**Q-function architecture.** Since large neural networks has been crucial for scaling to large, diverse datasets in NLP and vision (e.g., Tan & Le, 2019; Brown et al., 2020; Kaplan et al., 2020)), we explore using bigger architectures for scaling offline Q-learning. We use standard feature extractor backbones from vision, namely, the Impala-CNN architectures (Espeholt et al., 2018) that are fairly standard in deep RL and ResNet 34, 50 and 101 models from the ResNet family (He et al., 2016). We make modifications to these networks following recommendations from prior work (Kumar et al., 2022): we utilize group normalization instead of batch normalization in ResNets, and utilize point-wise multiplication with a learned spatial embedding when converting the output feature map of the vision backbone into a flattened vector which is to be fed into the feed-forward part of the Q-function.

To handle the multi-task setting, we use a multi-headed architecture where the Q-network outputs values for each game separately. The architecture uses a shared encoder and feedforward layers with separate linear projection layers for each game (Figure 3). The training objective (Eq. 1) is computed using the Q-values for the game that the transition originates from. In principle, explicitly injecting the task-identifier may be unnecessary and its impact could be investigated in future work.

**Feature Normalization via DR3 (Kumar et al., 2021a).** While the previous modifications lead to significant improvements over naïve CQL, our preliminary experiments on a subset of games did not attain good performance. In the single-task setting, Kumar et al. (2021a) proposes a regularizer that stabilizes training and allows the network to better use capacity, however, it introduces an additional hyperparameter to tune. Motivated by this approach, we regularize the magnitude of the learned features of the observation by introducing a "normalization" layer in the Q-network. This layer forces the learned features to have an $\ell_2$ norm of 1 by construction, and we found that this this speeds up learning, resulting in better performance. We present an ablation study analyzing this choice in Table 2. We found this sufficient to achieve strong performance, however, we leave exploring alternative feature normalization schemes to future work.

> **To summarize,** the primary modifications that enable us to scale CQL are: **(1)** use of large ResNets with learned spatial embeddings and group normalization, **(2)** use of a distributional representation of return values and cross-entropy loss for training (i.e., C51 (Bellemare et al., 2017)), and **(3)** feature normalization at intermediate layers to prevent feature co-adaptation, motivated by Kumar et al. (2021a). For brevity, we call our approach **Scaled Q-learning**.

## 5 EXPERIMENTAL EVALUATION

In our experiments, we study how our approach, scaled Q-learning, can simultaneously learn from sub-optimal and optimal data collected from 40 different Atari games. We compare the resulting multi-task policies to behavior cloning (BC) with same architecture as scaled QL, and the prior state-of-the-art method based on decision transformers (DT) (Chen et al., 2021), which utilize return-conditioned supervised learning with large transformers (Lee et al., 2022), and have been previously proposed for addressing this task. We also study the efficacy of the multi-task initialization produced by scaled Q-learning in facilitating rapid transfer to new games via both offline and online fine-tuning, in comparison to state-of-the-art self-supervised representation learning methods and other prior approaches. Our goal is to answer the following questions: **(1)** How do our proposed design decisions impact performance scaling with high-capacity models?, **(2)** Can scaled QL more effectively leverage higher model capacity compared to naïve instantiations of Q-learning?, **(3)** Do the representations

learned by scaled QL transfer to new games? We will answer these questions in detail through multiple experiments in the coming sections, but we will first summarize our main results below.

**Main empirical findings.** Our main results are summarized in Figures 2 and 5. These figures show the performance of scaled QL, multi-game decision transformers (Lee et al., 2022) (marked as "DT"), a prior method based on supervised learning via return conditioning, and standard behavioral cloning baselines (marked as "BC") in the two settings discussed previously, where we must learn from: (i) near optimal data, and (ii) sub-optimal data obtained from the initial 20% segment of the replay buffer (see Section 3 for problem setup). See Figure 4 for a direct comparison between DT and BC.

In the more challenging sub-optimal data setting, scaled QL attains a performance of **77.8%** IQM human-normalized score, although trajectories in the sub-optimal training dataset only attain 51% IQM human-normalized score. Scaled QL also outperforms the prior DT approach by **2.5 times** on this dataset, even though the DT model has more than twice as many parameters and uses data augmentation, compared to scaled QL.

In the $2^{nd}$ setting with near-optimal data, where the training dataset already contains expert trajectories, scaled QL with 80M parameters still outperforms the DT approach with 200M parameters, although the gap in performance is small (3% in IQM performance, and 20% on median performance). Overall, these results show that scaled QL is an effective approach for learning from large multi-task datasets, for a variety of

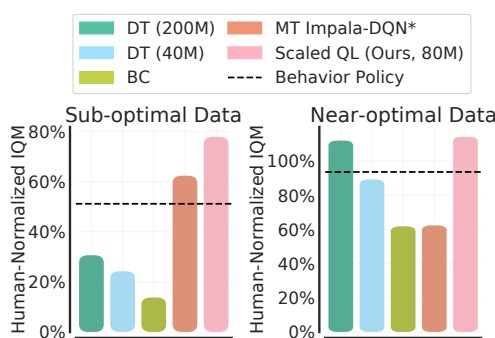

Figure 5: **Offline scaled conservative Q-learning vs other prior methods** with near-optimal data and sub-optimal data. Scaled QL outperforms the best DT model, attaining an IQM human-normalized score of **114.1%** on the near-optimal data and **77.8%** on the sub-optimal data, compared to 111.8% and 30.6% for DT, respectively.

data compositions including sub-optimal datasets, where we must stitch useful segments of suboptimal trajectories to perform well, and near-optimal datasets, where we should attempt to mimic the best behavior in the offline dataset.

To the best of our knowledge, these results represent the largest performance improvement over the average performance in the offline dataset on such a challenging problem. We will now present experiments that show that offline Q-learning scales and generalizes.

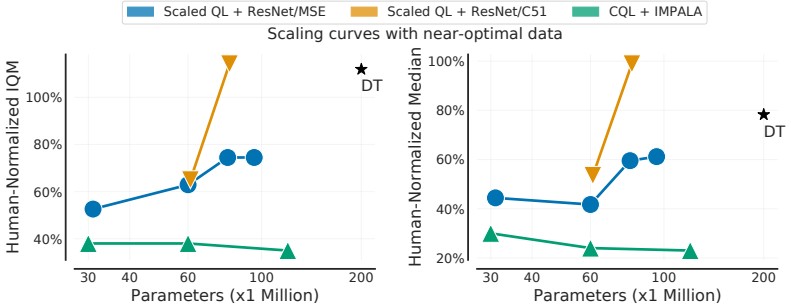

Figure 6: **Scaling trends for offline Q-learning.** Observe that while the performance of scaled QL instantiated with IMPALA architectures (Espeholt et al., 2018) degrades as we increase model size, the performance of scaled QL utilizing the ResNets described in Section 4 continues to increase as model capacity increases. This is true for both an MSE-style TD error as well as for the categorical TD error used by C51 (which performs better on an absolute scale). The CQL + IMPALA performance numbers are from (Lee et al., 2022).

## 5.1 DOES OFFLINE Q-LEARNING SCALE FAVORABLY?

One of the primary goals of this paper was to understand if scaled Q-learning is able to leverage the benefit of higher capacity architectures. Recently, Lee et al. (2022) found that the performance of CQL with the IMPALA architecture does not improve with larger model sizes and may even degrade with larger model sizes. To verify if scaled Q-learning can address this limitation, we compare our value-based offline RL approach with a variety of model families: **(a)** IMPALA family (Espeholt et al., 2018): three IMPALA models with varying widths $(4, 8, 16)$ whose performance numbers are

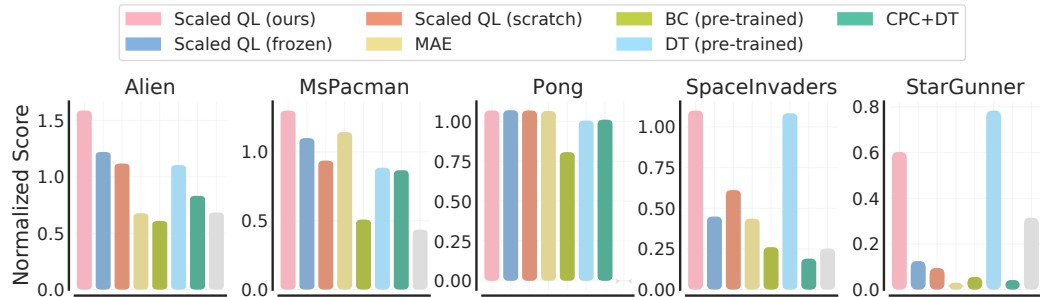

Figure 7: **Offline fine-tuning** performance on unseen games trained with 1% of held-out game's data, measured in terms of DQN-normalized score, following (Lee et al., 2022). On average, pre-training with scaled QL outperforms other methods by **82%**. Furthermore, scaled QL improves over scaled QL (scratch) by 45%, indicating that the representations learned by scaled QL during multi-game pre-training are useful for transfer. Self-supervised representation learning (CPC, MAE) alone does not attain good fine-tuning performance.

taken directly from Lee et al. (2022) (and was consistent with our preliminary experiments), **(b)** ResNet 34, 50, 101 and 152 from the ResNet family, modified to include group normalization and learned spatial embeddings.These architectures include both small and large networks, spanning a wide range from 1M to 100M parameters. As a point of reference, we use the scaling trends of the multi-game decision transformer and BC transformer approaches from Lee et al. (2022).

Observe in Figure 6 that the performance of scaled Q-learning improves as the underlying Q-function model size grows. Even though the standard mean-squared error formulation of TD error results in worse absolute performance than C51 (blue vs orange), for both of these versions, the performance of scaled Q-learning increases as the models become larger. This result indicates that value-based offline RL methods can scale favorably, and give rise to better results, but this requires carefully picking a model family. This also explains the findings from Lee et al. (2022): while this prior work observed that CQL with IMPALA scaled poorly as model size increases, they also observed that the performance of return-conditioned RL instantiated with IMPALA architectures also degraded with higher model sizes. Combined with the results in Figure 6 above, this suggests that poor scaling properties of offline RL can largely be attributed to the choice of IMPALA architectures, which may not work well in general even for supervised learning methods (like return-conditioned BC).

## 5.2 CAN OFFLINE RL LEARN USEFUL INITIALIZATIONS THAT ENABLE FINE-TUNING?

Next, we study how multi-task training on multiple games via scaled QL can learn general-purpose representations that can enable *rapid* fine-tuning to new games. We study this question in two scenarios: fine-tuning to a new game via offline RL with a small amount of held-out data (1% uniformly subsampled datasets from DQN-Replay (Agarwal et al., 2020)), and finetuning to a new game mode via sample-efficient online RL initialized from our multi-game offline Q-function. For finetuning, we transfer the weights from the visual encoder and reinitialize the downstream feed-forward component (Figure 1). For both of these scenarios, we utilize a ResNet101 Q-function trained via the methodology in Section 4, using C51 and feature normalization.

**Scenario 1 (Offline fine-tuning)**: First, we present the results for fine-tuning in an offline setting: following the protocol from Lee et al. (2022), we use the pre-trained representations to rapidly learn a policy for a novel game using limited offline data (1% of the experience of an online DQN run). In Figure 7, we present our results for offline fine-tuning on 5 games from Lee et al. (2022), ALIEN, MSPACMAN, SPACE INVADERS, STARGUNNER and PONG, alongside the prior approach based on decision transformers ("DT (pre-trained)"), and fine-tuning using pre-trained representations learned from state-of-the-art self-supervised representation learning methods such as contrastive predictive coding (CPC) (Oord et al., 2018) and masked autoencoders (MAE) (He et al., 2021). For CPC performance, we use the baseline reported in Lee et al. (2022). MAE is a more recent self-supervised approach that we find generally outperformed CPC in this comparison. For MAE, we first pretrained a vision transformer (ViT-Base) (Dosovitskiy et al., 2020) encoder with 80M parameters trained via a reconstruction loss on observations from multi-game Atari dataset and freeze the encoder weights as done in prior work (Xiao et al.). Then, with this frozen visual encoder, we used the same feed forward architecture, Q-function parameterization, and training objective (CQL with C51) as scaled QL to

Figure 8: **Online fine-tuning** results on unseen game *variants*. **Left**. The top row shows default variants and the bottom row shows unseen variants evaluated for transfer: Freeway's mode 1 adds buses, more vehicles, and increases velocity; Hero's mode 1 starts the agent at level 5; Breakout's mode 12 hides all bricks unless the ball has recently collided with a brick. **Right**. We fine-tune all methods except single-game DQN for 3M online frames (as we wish to test fast online adaptation). Error bars show minimum and maximum scores across 2 runs while the bar shows their average. Observe that scaled QL significantly outperforms learning from scratch and single-game DQN with 50M online frames. Furthermore, scaled QL also outperforms RL fine-tuning on representations learned using masked auto-encoders. See Figure B.1 for learning curves.

finetune the MAE network. We also compare to baseline methods that do not utilize any multi-game pre-training (DT (scratch) and Scaled QL (scratch)).

**Results.** Observe in Figure 7 that multi-game pre-training via scaled QL leads to the best fine-tuning performance and improves over prior methods, including decision transformers trained from scratch. Importantly, we observe *positive transfer* to new games via scaled QL. Prior works (Badia et al., 2020) running multi-game Atari (primarily in the online setting) have generally observed negative transfer across Atari games. We show for the first time that pre-trained representations from Q-learning enable positive transfer to novel games that significantly outperforms return-conditioned supervised learning methods and dedicated representation learning approaches.

**Scenario 2 (Online fine-tuning)**: Next, we study the efficacy of the learned representations in enabling online fine-tuning. While deep RL agents on ALE are typically trained on default game modes (referred to as $m0d0$), we utilize new *variants* of the ALE games designed to be challenging for humans (Machado et al., 2018) for online-finetuning. We investigate whether multi-task training on the 40 default game variants can enable fast online adaptation to these never-before-seen variants. In contrast to offline fine-tuning (Scenario 1), this setting tests whether scaled QL can also provide a good initialization for online data collection and learning, for closely related but different tasks. Following Farebrother et al. (2018), we use the same *variants* investigated in this prior work: BREAKOUT, HERO, and FREEWAY, which we visualize in Figure 8 (left). To disentangle the performance gains from multi-game pre-training and the choice of Q-function architecture, we compare to a baseline approach ("scaled QL (scratch)") that utilizes an identical Q-function architecture as pre-trained scaled QL, but starts from a random initialization. As before, we also evaluate fine-tuning performance using the representations obtained via masked auto-encoder pre-training (He et al., 2021; Xiao et al.). We also compare to a single-game DQN performance attained after training for 50M steps, $16\times$ more transitions than what is allowed for scaled QL, as reported by Farebrother et al. (2018).

**Results**. Observe in Figure 8 that fine-tuning from the multi-task initialization learned by scaled QL significantly outperforms training from scratch as well as the single-game DQN run trained with **16x** more data. Fine-tuning with the frozen representations learned by MAE performs poorly, which we hypothesize is due to differences in game dynamics and subtle changes in observations, which must be accurately accounted for in order to learn optimal behavior (Dean et al., 2022). Our results confirm that offline Q-learning can both effectively benefit from higher-capacity models and learn multi-task initializations that enable sample-efficient transfer to new games.

## 5.3 ABLATION STUDIES

Finally, in this section we perform controlled ablation studies to understand how crucial the design decisions introduced in Section 4 are for the success of scaled Q-learning. In particular, we will attempt to understand the benefits of using C51 and feature normalization.

**MSE vs C51:** We ran scaled Q-learning with identical network architectures (ResNet 50 and ResNet 101), with both the conventional squared error formulation of TD error, and compare it to C51, which our main results utilize. Observe in Table 1 that C51 leads to much better performance for both

Table 1: **Performance of Scaled QL with the standard mean-squared TD-error and C51** in the offline 40-game setting aggregated by the median human-normalized score. Observe that for both ResNet 50 and ResNet 101, utilizing C51 leads to a drastic improvement in performance.

|  | Scaled QL (ResNet 50) | Scaled QL (ResNet 101) |
|---|---|---|
| **with MSE** | 41.1% | 59.5% |
| **with C51** | 53.5% (+12.4%) | 98.9% (+39.4%) |

ResNet 50 and ResNet 101 models. The boost in performance is the largest for ResNet 101, where C51 improves by over **39%** as measured by median human-normalized score. This observation is surprising since prior work (Agarwal et al., 2021) has shown that C51 performs on par with standard DQN with an Adam optimizer, which all of our results use. One hypothesis is that this could be the case as TD gradients would depend on the scale of the reward function, and hence some games would likely exhibit a stronger contribution in the gradient. This is despite the fact that our implementation of MSE TD-error already attempts to correct for this issue by applying the unitary scaling technique from (Kurin et al., 2022) to standardize reward scales across games. That said, we still observe that C51 performs significantly better.

**Importance of feature normalization:** We ran small-scale experiments with and without feature normalization (Section 4). In these experiments, we consider a multi-game setting with only 6 games: ASTERIX, BREAKOUT, PONG, SPACEINVADERS, SEAQUEST, and we train with the initial 20% data for each game. We report aggregated median human-normalized score across the 6 games in Table 2 for three different network architectures (ResNet 34, ResNet 50 and ResNet 101). Observe that the addition of feature normalization significantly improves performance for all the models. Motivated by this initial empirical finding, we used feature normalization in all of our main experiments. Overall, the above ablation studies validate the efficacy of the two key design decisions in this paper.

Table 2: **Performance of Scaled QL with and without feature normalization in the 6 game setting** reported in terms of the median human-normalized score. Observe that with models of all sizes, the addition of feature normalization improves performance.

|  | Scaled QL (ResNet 34) | Scaled QL (ResNet 50) | Scaled QL (ResNet 101) |
|---|---|---|---|
| **without feature normalization** | 50.9% | 73.9% | 80.4% |
| **with feature normalization** | 78.0% (+28.9%) | 83.5% (+9.6%) | 98.0% (+17.6%) |

**Additional ablations:** We also conducted ablation studies for the choice of the backbone architecture (spatial learned embeddings) in Appendix B.3, and observed that utilizing spatial embeddings is better. We also evaluated the performance of scaled QL without conservatism to test the importance of utilizing pessimism in our setting with diverse data in Appendix B.4, and observe that pessimism is crucial for attaining good performance on an average. We also provide some scaling studies for another offline RL method (discrete BCQ) in Appendix B.2.

## 6 DISCUSSION

This work shows, for the first time, that offline Q-learning can scale to high-capacity models trained on large, diverse datasets. As we hoped, by scaling up capacity, we unlocked analogous trends to those observed in vision and NLP. We found that scaled QL trains policies that exceed the average dataset performance and prior methods, especially when the dataset does not contain expert trajectories. Furthermore, by training a large-capacity model on diverse tasks, we show that Q-learning is sufficient to recover general-purpose representations that enable rapid learning of novel tasks. Although we detailed an approach that is sufficient to scale Q-learning, this is by no means optimal. The scale of the experiments limited the number of alternatives we could explore, and we expect that future work will greatly improve performance. For example, contrary to DT (Lee et al., 2022), we did not use data augmentation in our experiments, which we believe can provide significant benefits. While we did a preliminary attempt to perform online fine-tuning on an entirely new game (SPACEINVADERS), we found that this did not work well for any of the pretrained representations (see Figure B.1). Addressing this is an important direction for future work. We speculate that this challenge is related to designing methods for learning better exploration from offline data, which is not required for offline fine-tuning. We refer the reader to the appendix for further discussion of future work. Overall, we believe that scaled QL could serve as a starting point for RL-trained foundation models.

AUTHOR CONTRIBUTIONS

AK conceived and led the project, developed scaled QL, decided and ran most of the experiments. RA discussed the experiment design and project direction, helped set up and debug the training pipeline, took the lead on setting up and running the MAE baseline and the online fine-tuning experiments. XG helped with design choices for some experiments. GT advised the project and ran baseline DT experiments. SL advised the project and provided valuable suggestions. AK, RA, GT, SL all contributed to writing and editing the paper.

ACKNOWLEDGEMENTS

We thank several members of the Google Brain team for their help, support and feedback on this paper. We thank Dale Schuurmans, Dibya Ghosh, Ross Goroshin, Marc Bellemare and Aleksandra Faust for informative discussions. We thank Sherry Yang, Ofir Nachum, and Kuang-Huei Lee for help with the multi-game decision transformer codebase; Anurag Arnab for help with the Scenic ViT codebase. We thank Zoubin Ghahramani and Douglas Eck for leadership support.

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

# Appendices

## A    FUTURE WORK

In addition to future work discussed in Section 6, here we list other promising directions for future work. Given the strong performance of transformers, we suspect that offline Q-learning with a transformer architecture is a promising future direction. Furthermore, there are several avenues for future investigation related to the ablation studies in Section 5.3 for key design choices in scaled Q-learning: 1) it is unclear if C51 works better because of the distributional formulation or the categorical representation and experiments with other distributional formulations could answer this question, 2) we did not extensively try alternate feature normalization schemes, which may improve results. Another important avenue for future work is to scale offline Q-learning on other RL domains such as robotic navigation, manipulation, locomotion, education, etc. This would require building large-scale tasks, and we believe that scaled QL would provide for a good starting point for scaling in these domains. Finally, in line with Agarwal et al. (2022), we plan to release our pre-trained models, which we hope would enable subsequent methods to build upon.

## B    ADDITIONAL RESULTS

### B.1    ADDITIONAL RESULTS FROM THE PAPER

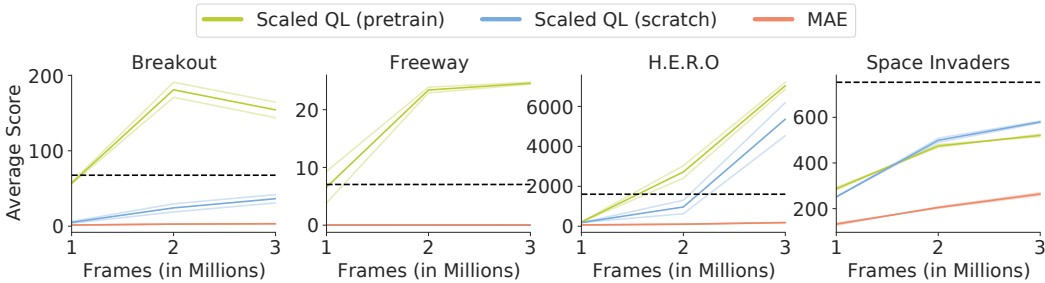

Figure B.1: Learning curves for **online fine-tuning on unseen game variants**. The dotted horizontal line shows the performance of a single-game DQN agent trained for 50M frames (16x more data than our methods). See Figure 8 for visualization of the variants.

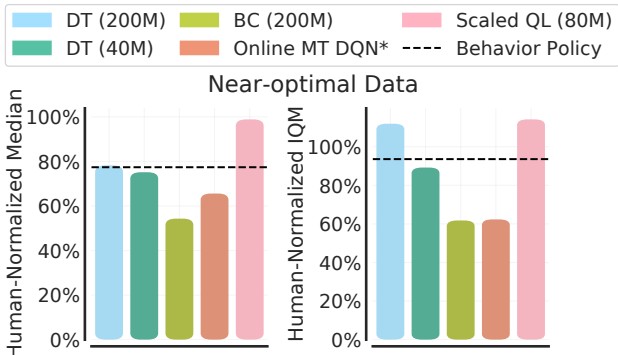

Figure B.2: **Offline scaled conservative Q-learning vs other prior methods** with near-optimal data. Scaled QL outperforms the best DT model, attaining an IQM human-normalized score of **114.1%** and a median human-normalized score of **98.9%** compared to 111.8% and 78.2% for DT, respectively.

### B.2    RESULTS FOR SCALING DISCRETE-BCQ

To implement discrete BCQ, we followed the official implementation from Fujimoto et al. (2019). We first trained a model of the behavior policy, $\widehat{\pi}_\beta(\mathbf{a}|\mathbf{s})$, using an architecture identical to that of the

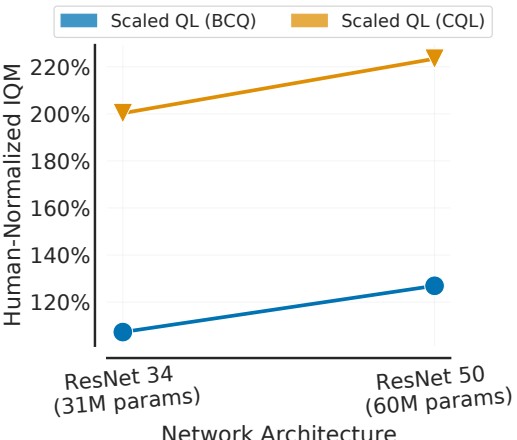

Figure B.3: **Performance of scaling CQL and BCQ in terms of IQM human-normalized score.** We perform this comparison on the six-game setting for 100 epochs (note that these results are after 2x longer training than other ablations in Table 2). Observe that for discrete-BCQ the performance improves from ResNet 34 to ResNet 50, indicating that it does scale favorably as network capacity increases.

Q-function, using negative log-likelihood. Then, following Fujimoto et al. (2019), we updated the Bellman backup to only perform the maximization over actions that attain a high likelihood under the probabilities learned by the behavior policy, as shown below:

$$y(\mathbf{s}, \mathbf{a}) := r(\mathbf{s}, \mathbf{a}) + \gamma \max_{\mathbf{a}':\widehat{\pi}_\beta(\mathbf{a}'|\mathbf{s}') \geq \tau \cdot \max_{\mathbf{a}''} \widehat{\pi}_\beta(\mathbf{a}''|\mathbf{s}')} \bar{Q}(\mathbf{s}', \mathbf{a}'),$$

where $\tau$ is a hyperparameter. To tune the value of $\tau$, we ran a preliminary initial sweep over $\tau = \{0.05, 0.1, 0.3\}$. When using C51 in our setup, we had to use a smaller CQL $\alpha$ of 0.05 (instead of 0.1 for the MSE setting from Kumar et al. (2021a)), possibly because a discrete representation of Q-values used by C51 is less prone to overestimation. Therefore, in the case of discrete-BCQ, we chose to perform an initial sweep over $\tau$ values that were smaller than or equal to (i.e., less conservative) the value of $\tau = 0.3$ used in Fujimoto et al. (2019).

Since BCQ requires an additional policy network, it imposes a substantial memory overhead and as such, we performed a sweep for initial 20 iterations to pick the best $\tau$. We found that in these initial experiments, $\tau = 0.05$ performed significantly worse, but $\tau = 0.1$ and $\tau = 0.3$ performed similarly. So, we utilized $\tau = 0.3$ for reporting these results.

We ran these scaling experiments with ResNet 34 and ResNet 50 in the six-game setting and report human-normalized IQM performance after 100 epochs = 6.25M gradient steps in Figure B.3. We also present the results for CQL on the side for comparisons. Observe that we find favorable scaling trends for BCQ: average performance over all games increases as the network size increases, indicating that other offline RL algorithms such as BCQ can scale as we increase network capacity.

### B.3 ABLATION FOR BACKBONE ARCHITECTURE

In this section, we present some results ablating the choice of the backbone architecture. For this ablation, we ablate the choice of the spatial embedding while keeping group normalization fixed in both cases. We perform this study for the 40-game setting. Observe that using the learned spatial embedding results in better performance, and improves in 27 out of 40 games compared to not using the learned embeddings.

Regarding the choice of group normalization vs batch normalization, note that we have been operating in a setting where the size of the batch per device / core is only 4. Particularly, we use Cloud TPU v3 accelerators with 64 / 128 cores, and bigger batch sizes than 4 do not fit in memory, especially for larger-capacity ResNets. This means that if we utilized batch normalization, we would be computing batch statistics over only 4 elements, which is known to be unstable even for standard computer vision tasks, for example, see Figure 1 in Wu & He (2018).

Table B.1: **Ablations for the backbone architecture in the 40-game setting** with ResNet 101. Observe that learned spatial embeddings leads to around 80% improvement in performance.

|  | Scaled QL without backbone | Scaled QL w/ backbone |
|---|---|---|
| **Median human-normalized score** | 54.9% | **98.9**% |
| **IQM human-normalized score** | 68.9% | **114.1**% |
| **Num. games with better performance** | 13 / 40 | **27 / 40** |

### B.4 RESULTS FOR SCALED QL WITHOUT PESSIMISM

In Table B.2, we present the results of running scaled Q-learning with no conservatism, i.e., by setting the value of $\alpha$ in Equation 1 to 0.0 in the six game setting. We utilize the entire DQN-replay dataset (Agarwal et al., 2020) for each of these six games that would be present in the full 40-game dataset, to preserve the per-game dataset diversity.

Observe that while utilizing no conservatism does still learn, the performance of scaled QL without conservatism is notably worse than standard scaled QL. Interestingly, on ASTERIX, the performance without pessimism is better than performance with pessimism, whereas, the use of pessimism in SPACEINVADERS and SEAQUEST leads to at least 2x improvement in performance.

Table B.2: **Performance of scaled QL with and without conservatism in terms of IQM human-normalized score** in the six-game setting for 100 epochs (2x longer training compared to other ablations in Table 2) performed with a ResNet 50. Observe that utilizing conservatism via CQL is beneficial. We also present per-game raw scores in this table. Observe that while in one games no pessimism with such data can outperform CQL, we do find that overall, conservatism performs better.

|  | Scaled QL without CQL | Scaled QL w/ CQL |
|---|---|---|
| ASTERIX | 38000 | 35200 |
| BREAKOUT | 322 | 410 |
| PONG | 12.6 | 19.8 |
| QBERT | 13800 | 15500 |
| SEAQUEST | 1378 | 3694 |
| SPACEINVADERS | 1675 | 3819 |
| **IQM human-normalized score** | 188.3% | **223.4**% |

We also present some results without pessimism in the complete 40-game setting in Table B.3. Unlike the smaller six game setting, we find a much larger difference between no pessimism (without CQL) and utilizing pessimism via CQL. In particular, we find that in 6 games, not using pessimism leads to slightly better performance, but this strategy hurts in all other games, giving rise to an agent that performs worse than random in many of these 34 games. This indicates that pessimism is especially deisrable as the diversity of tasks increases.

Table B.3: **Scaled QL with and without conservatism in terms of IQM human-normalized score in the 40-game setting** with ResNet 101. Observe that utilizing conservatism via CQL is still beneficial.

|  | Scaled QL without CQL | Scaled QL w/ CQL |
|---|---|---|
| **Median human-normalized score** | 11.1% | 98.9% |
| **IQM human-normalized score** | 13.5% | 114.1% |
| **Num. games with better performance** | 6 / 40 | **34 / 40** |

## C IMPLEMENTATION DETAILS AND HYPER-PARAMETERS

In this section, we will describe some of the implementation details behind our method and will provide implementation details for our approach, including the details of the network architectures, the details of feature normalization and the details of our training and evaluation protocol.

## C.1 NETWORK ARCHITECTURE

In our primary experiments, we consider variants of ResNet architectures for scaled Q-Learning. The vision backbone in these architectures mimic the corresponding ResNet architectures from He et al. (2016), however, we utilize group normalization (Wu & He, 2018) (with a group size of 4) instead of batch normalization, and instead of applying global mean pooling to aggregate the outputs of the ResNet, we utilize learned spatial embeddings (Kumar et al., 2022), that learn a matrix that point-wise multiplies the output feature map of the ResNet. The output volume is then flattened to be passed as input to the feed-forward part of the network.

The feed-forward layer part of the network begins with a layer of size 2048, and then applies layer norm on the network. After this we apply 3 feed-forward layers with hidden dimension 1024 with ReLU activations, to obtain the representation of the image observation.

Then, we apply feature normalization to the representation, by applying a normalization layer which divides the representation of a given observation by its $\ell_2$ norm. Note that we do pass gradients through this normalization term. Now, this representation is passed into different heads that are supposed to predict the Q-values. The total number of heads is equal to the number of games we train on. Each head consists of a linear layer that maps the 1024-dimensional normalized representation to a vector of $K$ elements, where $K = |\mathcal{A}|$ (i.e., the size of the action space) for the standard real-valued parameterization of Q-values, and $K = |\mathcal{A}| \times 51$ for C51. The network does not apply any output activation in either case, and the Q-values are treated as logits for C51.

## C.2 DETAILS OF C51

For the main results in the paper, we utilize C51. The main hyperparameter in C51 is the size of the support set of Q-values. Unlike the paper from Bellemare et al. (2017) which utilizes a support set of $[-10, 10]$, we utilize a support set of $[-20, 20]$ to allow for flexibility of CQL: Applying the CQL regularizer can underestimate or overestimate Q-values, and this additional flexibility aids such scenarios. Though, we still utilize only 51 atoms in our support set, and the average dataset Q-value in our training runs is generally always smaller, around $\sim 8 - 9$.

## C.3 TRAINING AND EVALUATION PROTOCOLS AND HYPERPARAMETERS

We utilize the initial 20% (sub-optimal) and 100% (near-optimal) datasets from Agarwal et al. (2020) for our experiments. These datasets are generated from runs of standard online DQN on stochastic dynamics Atari environments that utilize sticky actions, *i.e.*, there is a 25% chance at every time step that the environment will execute the agents previous action again, instead of the new action commanded. The majority of the training details are identical to a typical run of offline RL on single-game Atari. We discuss the key differences below.

We trained our ResNet 101 network for $10M$ gradient steps with a batch size of 512. The agent hasn't converged yet, and the performance is still improving gradually. When training on multiple games, we utilize a stratified batch sampling scheme with a total batch size of $512$. To obtain the batch at any given training iteration, we first sample 128 game indices from the set all games (40 games in our experiments) with replacement, and then sample 4 transitions from each game. This scheme does not necessarily produce an equal number of transitions from each game in a training batch, but it does make sure that all games are seen in expectation throughout training.

Since we utilize a larger batch size, that is 16 times larger than the standard batch size of 32 on Atari, we scale up the learning rate from $5e - 05$ to $0.0002$, but keep the target network update period fixed to the same value of 1 target update per 2000 gradient steps as with single-task Atari. We also utilize $n$-step returns with $n = 3$ by default, with both our MSE and C51 runs.

**Evaluation Protocol.** Even though we train on Atari datasets with sticky actions, we evaluate on Atari environments that do not enable sticky actions following the protocol from Lee et al. (2022). This allows us to be comparable to this prior work in all of our comparisons, without needing to re-train their model, which would have been too computationally expensive. Following standard protocols on Atari, we evaluate a noised version of the policy with an epsilon-greedy scheme, with $\varepsilon_{\text{eval}} = 0.001$. Following the protocol in Castro et al. (2018), we compute average return over 125K training steps.

Table C.1: **Hyperparameters used by multi-game training.** Here we report the key hyperparameters used by the multi-game training. The differences from the standard single-game training setup are highlighted in red.

| Hyperparameter | Setting (for both variations) |
|---|---|
| Eval Sticky actions | No |
| Grey-scaling | True |
| Observation down-sampling | (84, 84) |
| Frames stacked | 4 |
| Frame skip (Action repetitions) | 4 |
| Reward clipping | [-1, 1] |
| Terminal condition | Game Over |
| Max frames per episode | 108K |
| Discount factor | 0.99 |
| Mini-batch size | 512 |
| Target network update period | every 2000 updates |
| Training environment steps per iteration | 62.5k |
| Update period every | 1 environment steps |
| Evaluation $\epsilon$ | 0.001 |
| Evaluation steps per iteration | 125K |
| Learning rate | 0.0002 |
| n-step returns ($n$) | 3 |
| CQL regularizer weight $\alpha$ | 0.1 for MSE, 0.05 for C51 |

## C.4 FINE-TUNING PROTOCOL

**For offline fine-tuning** we fine-tuned the parameters of the pre-trained policy on the new domain using a batch size of 32, and identical hyperparameters as those used during pre-training. We utilized $\alpha = 0.05$ for fine-tuning, but with the default learning rate of $5e - 05$ (since the batch size was the default 32). We attempted to use other CQL $\alpha$ values $\{0.07, 0.02, 0.1\}$ for fine-tuning but found that retaining the value of $\alpha = 0.05$ for pre-training worked best. For reporting results, we reported the performance of the algorithm at the end of 300k gradient steps.

**For online fine-tuning**, we use the C51 algorithm (Bellemare et al., 2017), with $n$-step= 3 and all other hyperparameters from the C51 implementation in the Dopamine library (Castro et al., 2018). We swept over two learning rates, $\{1e - 05, 5e - 05\}$ for all the methods and picked the best learning rate per-game for all the methods. For the MAE implementation, we used the Scenic library (Dehghani et al., 2022) with the typical configuration used for ImageNet pretraining, except using $84 \times 84 \times 4$ sized Atari observations, instead of images of size $224 \times 224 \times 3$. We train the MAE for 2 epochs on the entire multi-task offline Atari dataset and we observe that the reconstruction loss plateaus to a low value.

## C.5 DETAILS OF MULTI-TASK IMPALA DQN

The "MT Impala DQN" comparison in Figures 2 & 5 is a multi-task implementation of online DQN, evaluated at 5x many gradient steps as the size of the sub-optimal dataset. This comparison is taken directly from Lee et al. (2022). To explain this baseline briefly, this baseline runs C51 in conjunction with n-step returns with $n = 4$, with an IMPALA architecture that uses three blocks with 64, 128, and 128 channels. This baseline was trained with a batch size of 128 and update period of 256.

## D RAW TRAINING SCORES FOR DIFFERENT MODELS

Table D.1: Raw scores on 40 training Atari games in the sub-optimal multi-task Atari dataset (51% human-normalized IQM). Scaled QL uses the ResNet-101 architecture.

| Game | DT (200M) | DT (40M) | Scaled QL (80M) | BC (80M) | MT Impala-DQN* | Human |
|---|---|---|---|---|---|---|
| Amidar | 72.9 | 82.2 | 33.1 | 14.5 | 629.8 | 1719.5 |
| Assault | 392.9 | 124.7 | 1380.8 | 1060.0 | 1338.7 | 742.0 |
| Asterix | 1518.8 | 2256.2 | 9967.3 | 745.3 | 2949.1 | 8503.3 |
| Atlantis | 10525.0 | 13125.0 | 485200.0 | 2494.1 | 976030.4 | 29028.1 |
| BankHeist | 13.1 | 15.6 | 18.6 | 87.6 | 1069.6 | 753.1 |
| BattleZone | 3750.0 | 7687.5 | 8500.0 | 1550.0 | 26235.2 | 37187.5 |
| BeamRider | 1535.8 | 1397.5 | 5856.5 | 327.2 | 1524.8 | 16926.5 |
| Boxing | 71.4 | 74.2 | 95.2 | 95.4 | 68.3 | 12.1 |
| Breakout | 38.8 | 38.2 | 351.1 | 274.7 | 32.6 | 30.5 |
| Carnival | 993.8 | 791.2 | 199.3 | 792.7 | 2021.2 | 3800.0 |
| Centipede | 2645.4 | 3026.9 | 2711.4 | 2260.8 | 4848.0 | 12017.0 |
| ChopperCommand | 1006.2 | 1093.8 | 752.2 | 336.7 | 951.4 | 7387.8 |
| CrazyClimber | 85487.5 | 86050.0 | 122933.3 | 121394.4 | 146362.5 | 35829.4 |
| DemonAttack | 2269.7 | 1049.4 | 14229.8 | 765.2 | 446.8 | 1971.0 |
| DoubleDunk | -14.5 | -20.2 | -12.4 | -13.6 | -156.2 | -16.4 |
| Enduro | 336.5 | 266.2 | 2297.6 | 638.7 | 896.3 | 860.5 |
| FishingDerby | 15.9 | 16.8 | 13.7 | -88.1 | -152.3 | -38.7 |
| Freeway | 16.2 | 20.5 | 24.4 | 0.1 | 30.6 | 29.6 |
| Frostbite | 1014.4 | 776.2 | 2324.5 | 234.8 | 2748.4 | 4334.7 |
| Gopher | 1137.5 | 1251.2 | 1041.0 | 231.5 | 3205.6 | 2412.5 |
| Gravitar | 237.5 | 193.8 | 260.3 | 248.8 | 492.5 | 3351.4 |
| Hero | 6741.2 | 6295.3 | 4011.9 | 7485.8 | 26568.8 | 30826.4 |
| IceHockey | -8.8 | -11.1 | -3.7 | -10.8 | -10.4 | 0.9 |
| Jamesbond | 378.1 | 312.5 | 58.7 | 7.1 | 264.6 | 302.8 |
| Kangaroo | 1975.0 | 2687.5 | 5796.6 | 307.1 | 7997.1 | 3035.0 |
| Krull | 6913.8 | 4377.5 | 9333.7 | 9585.3 | 8221.4 | 2665.5 |
| KungFuMaster | 17575.0 | 14743.8 | 24320.0 | 15778.6 | 29383.1 | 22736.3 |
| NameThisGame | 4396.9 | 4502.5 | 6759.6 | 2756.8 | 6548.8 | 8049.0 |
| Phoenix | 3560.0 | 2813.8 | 12770.0 | 762.9 | 3932.5 | 7242.6 |
| Pooyan | 1053.8 | 1394.7 | 1264.5 | 718.7 | 4000.0 | 4000.0 |
| Qbert | 8371.9 | 5917.2 | 14877.9 | 5759.6 | 4226.5 | 13455.0 |
| Riverraid | 6191.9 | 4265.6 | 9602.7 | 6657.2 | 7306.6 | 17118.0 |
| Robotank | 14.9 | 12.8 | 17.4 | 5.7 | 9.2 | 11.9 |
| Seaquest | 781.9 | 512.5 | 1021.8 | 113.9 | 1415.2 | 42054.7 |
| TimePilot | 2512.5 | 2700.0 | 767.3 | 3841.1 | -883.1 | 5229.2 |
| UpNDown | 5288.8 | 5456.2 | 35541.3 | 8395.2 | 8167.6 | 11693.2 |
| VideoPinball | 1277.4 | 1953.1 | 40.0 | 2650.3 | 85351.0 | 17667.9 |
| WizardOfWor | 237.5 | 881.2 | 107.0 | 495.3 | 975.9 | 4756.5 |
| YarsRevenge | 11867.4 | 10436.8 | 11482.4 | 17755.5 | 18889.5 | 54576.9 |
| Zaxxon | 287.5 | 337.5 | 1.4 | 0.0 | -0.1 | 9173.3 |

Table D.2: Raw scores on 40 training Atari games in the near-optimal multi-task Atari dataset. Scaled QL uses the ResNet 101 architecture.

| Game | DT (200 M) | DT (40M) | BC (200M) | MT Impala-DQN* | Scaled QL (80M) | Human |
|---|---|---|---|---|---|---|
| Amidar | 101.5 | 1703.8 | 101.0 | 629.8 | 21.0 | 1719.5 |
| Assault | 2385.9 | 1772.2 | 1872.1 | 1338.7 | 3809.6 | 742.0 |
| Asterix | 14706.3 | 4575.0 | 5162.5 | 2949.1 | 34278.9 | 8503.3 |
| Atlantis | 3105342.3 | 304931.2 | 4237.5 | 976030.4 | 881980.0 | 29028.1 |
| BankHeist | 5.0 | 40.0 | 63.1 | 1069.6 | 33.9 | 753.1 |
| BattleZone | 17687.5 | 17250.0 | 9250.0 | 26235.2 | 8812.5 | 37187.5 |
| BeamRider | 8560.5 | 3225.5 | 4948.4 | 1524.8 | 10301.0 | 16926.5 |
| Boxing | 95.1 | 92.1 | 90.9 | 68.3 | 99.5 | 12.1 |
| Breakout | 290.6 | 160.0 | 185.6 | 32.6 | 415.0 | 30.5 |
| Carnival | 2213.8 | 3786.9 | 2986.9 | 2021.2 | 926.1 | 3800.0 |
| Centipede | 2463.0 | 2867.5 | 2262.8 | 4848.0 | 3168.2 | 12017.0 |
| ChopperCommand | 4268.8 | 3337.5 | 1800.0 | 951.4 | 832.2 | 7387.8 |
| CrazyClimber | 126018.8 | 113425.0 | 123350.0 | 146362.5 | 140500.0 | 35829.4 |
| DemonAttack | 23768.4 | 3629.4 | 7870.6 | 446.8 | 56318.3 | 1971.0 |
| DoubleDunk | -10.6 | -12.5 | -1.5 | -156.2 | -13.1 | -16.4 |
| Enduro | 1092.6 | 770.8 | 793.2 | 896.3 | 2345.8 | 860.5 |
| FishingDerby | 11.8 | 19.2 | 5.6 | -152.3 | 23.8 | -38.7 |
| Freeway | 30.4 | 32.8 | 29.8 | 30.6 | 31.9 | 29.6 |
| Frostbite | 2435.6 | 934.4 | 782.5 | 2748.4 | 3566.4 | 4334.7 |
| Gopher | 9935.0 | 3827.5 | 3496.2 | 3205.6 | 3776.9 | 2412.5 |
| Gravitar | 59.4 | 75.0 | 12.5 | 492.5 | 262.3 | 3351.4 |
| Hero | 20408.8 | 19667.2 | 13850.0 | 26568.8 | 20470.6 | 30826.4 |
| IceHockey | -10.1 | -5.2 | -8.3 | -10.4 | -1.5 | 0.9 |
| Jamesbond | 700.0 | 712.5 | 431.2 | 264.6 | 483.6 | 302.8 |
| Kangaroo | 12700.0 | 11581.2 | 12143.8 | 7997.1 | 2738.6 | 3035.0 |
| Krull | 8685.6 | 8295.6 | 8058.8 | 8221.4 | 10176.9 | 2665.5 |
| KungFuMaster | 15562.5 | 16387.5 | 4362.5 | 29383.1 | 25808.3 | 22736.3 |
| NameThisGame | 9056.9 | 7777.5 | 7241.9 | 6548.8 | 11647.0 | 8049.0 |
| Phoenix | 5295.6 | 4744.4 | 4326.9 | 3932.5 | 5264.0 | 7242.6 |
| Pooyan | 2859.1 | 1191.9 | 1677.2 | 4000.0 | 2020.1 | 4000.0 |
| Qbert | 13734.4 | 12534.4 | 11276.6 | 4226.5 | 15946.0 | 13455.0 |
| Riverraid | 14755.6 | 11330.6 | 9816.2 | 7306.6 | 18494.8 | 17118.0 |
| Robotank | 63.2 | 50.9 | 44.6 | 9.2 | 53.2 | 11.9 |
| Seaquest | 5173.8 | 3112.5 | 1175.6 | 1415.2 | 414.1 | 42054.7 |
| TimePilot | 2743.8 | 3487.5 | 1312.5 | -883.1 | 4220.5 | 5229.2 |
| UpNDown | 16291.3 | 9306.9 | 10454.4 | 8167.6 | 55512.9 | 11693.2 |
| VideoPinball | 1007.7 | 9671.4 | 1140.8 | 85351.0 | 285.7 | 17667.9 |
| WizardOfWor | 187.5 | 687.5 | 443.8 | 975.9 | 301.6 | 4756.5 |
| YarsRevenge | 28897.9 | 25306.3 | 20738.9 | 18889.5 | 24393.9 | 54576.9 |
| Zaxxon | 275.0 | 4637.5 | 50.0 | -0.1 | 2.1 | 9173.3 |

