# OpenReview forum: "Offline Q-learning on Diverse Multi-Task Data Both Scales And Generalizes"
_ICLR.cc/2023/Conference — ICLR 2023 notable top 5%_

### Official Review · Reviewer_k6mY · 2022-10-24

**Confidence:** 5
**Correctness:** 4
**Technical Novelty And Significance:** 3
**Empirical Novelty And Significance:** 3
**Recommendation:** 8

**Clarity, Quality, Novelty And Reproducibility:**

The paper is very well-written and structured, easy to follow and the evaluation and technical implementation are sound. No codebase has been provided to judge reproducibility. While the individual components that are being introduced are not necessarily novel in themselves, it is the combination and claimed model performance that is novel and important for the field.


**Strength And Weaknesses:**

Strengths:
- The paper shows that augmenting CQL by only three, rather simple but broadly applicable design choices enables learning a much improved single multi-task policy from suboptimal expert data.
- The method shows a nice scaling behaviour for models of increasing capacity
- Finetuning and adaptation experiments to new game variants are very compelling
- Convincing and superior experimental results over baseline
- Ablation studies to assess the relevance of each model choice.
- The paper is well-structured and easy to follow.
- Nice presentation of results

Weaknesses:
- The paper makes a reference to an anonymous work from which the backbone architecture seems to be motivated. Unfortunately, I could not find this work in the supplementary material to gain further insights into the design decisions and the importance of this component.
- The importance of the backbone architecture is unclear as there are missing ablation studies.
- Minor: the policy still necessitates individual heads for each game and it would have been interesting to see how limiting this particular design choice is for obtaining the reported performances.

Minor:
- p.5: “[...] we found that this this speeds […]”



**Summary Of The Paper:**

This paper presents a combination of learning techniques and model design choices that enable the training of a single policy in offline Q-learning settings that can generalize across tasks. The authors present extensive evaluations on the popular multi-task Atari test bed involving 40 different games. Authors show that the performance of their approach scales with capacity and even works when trained on suboptimal offline datasets. Lastly, models trained on diverse offline data show very favourable and fast adaptation to new game variants.

**Summary Of The Review:**

Progress in offline RL is still often lagging behind online RL and I believe this paper represents an important step forward by augmenting CQL by a set of important techniques and model choices that allow the training of a single multi-task RL policy from suboptimal datasets with very favourable scaling and adaptation properties. The authors present very convincing and strong empirical evidence on the very challenging Atari test bed. I, therefore, recommend acceptance of this paper.

---

> ### Author Response · Authors · 2022-11-13
> **Author Response: Added experimental studies**
>
> Thank you for your detailed comments, and for a positive assessment of our work! We are glad that you found our paper easy to follow, well-written and our results to be strong. To address your concerns, we have added an ablation study to validate the efficacy of our backbone architecture (**Appendix A.3**) and included our code in the supplementary material. Our detailed response follows:
>
> **We hope that most of your concerns have been addressed and, if so, you would consider updating your score. We’d be happy to engage in further discussions.**
> ____
>
> > **Missing ablation for backbone architecture**
>
> We ran the ablation for whether to utilize the learned spatial embeddings in the multi-game Atari setting on 40 games and present some preliminary results (only 60 epochs into training). These results are shown in **Table A.1** in **Appendix A.3** and copied them below. Observe that utilizing the spatial learned embeddings leads to better performance, and performs better on 33 out of 40 games.
>
> |                                            | **without spatial embeddings**| **with spatial embeddings** |
> |--------------------------------------------|-------------------------------------|--------------------------------|
> | **Median human-normalized score**     | 35.4%                              | 79.9%                         |
> | **IQM human-normalized score**       | 40.0%                              | 91.1%                         |
> | **Number of games with better perf.** | 7/40                                | 33/40                          |
>
> Regarding the choice of group normalization vs batch normalization, note that we have been operating in a setting where the size of the batch per device / core is only 4. This means that if we utilized batch normalization, we would be computing batch statistics over only 4 elements, which is known to be unstable even for standard computer vision tasks (such as Imagenet classification, https://arxiv.org/pdf/1803.08494.pdf). Therefore, we used group normalization.
>
> ___
>
> > **No codebase has been provided to judge reproducibility**
>
> We have now updated the supplementary zip to include preliminary code for our main experiments. We would clean up the remaining code too, and include it in the final version.
>
> ___
>
> > **Anonymous paper missing in supplementary**
>
> We apologize for missing to add the anonymous paper we based this architecture off. We have now updated the supplementary to add that paper, which also consists of an ablation of the backbone architecture. However, we have also added that ablation above for our setting now.
> ___
>
> > **Minor: the policy still necessitates individual heads for each game**
>
> In our preliminary experiments, we observed that with the MSE TD-error, having a single head for the Q-function (which is also the policy for discrete Atari) led to poor performance as the Q-values for different games have different ranges. We were able to improve this performance by utilizing separate heads for the Q-function for the MSE, and we kept this design choice for scaled QL.

---

> > ### Comment · Reviewer_k6mY · 2022-11-17
> > **Thank you for addressing my concerns; increasing confidence**
> >
> > I thank the reviewers for clarifying all of my remaining concerns and would like to express my appreciation for running additional ablations on the backbone and including the preliminary codebase. I am therefore happy to increase my confidence accordingly. As a minor note, I would encourage the authors to disclose the required computational cost and details to reproduce these experiments in the final revision.

---

> > > ### Author Response · Authors · 2022-11-18
> > > **Thank You!**
> > >
> > > Thank you for increasing your confidence. We will make sure to add a section discussing the computational cost and other details in the final version.

---

### Official Review · Reviewer_oQ8T · 2022-10-24

**Confidence:** 3
**Correctness:** 3
**Technical Novelty And Significance:** 3
**Empirical Novelty And Significance:** 3
**Recommendation:** 6

**Clarity, Quality, Novelty And Reproducibility:**

See above


**Strength And Weaknesses:**

## Paper strengths and contributions
**Motivation and intuition**
The motivation for addressing the problem of learning general-purpose representations in offline Q-learning is convincing.

**Novelty**
In my opinion, studying extrapolation beyond datasets even when trained entirely on a large but highly suboptimal dataset in order to increase model performance when the model capacity increases seem novel.

**Technical contribution**
- The idea of utilizing standard feature extractor backbones from vision (i.e., the Impala-CNN architectures) to improve performance as model capacity increases is intuitive and convincing.
- The outcomes of combining and empirically verifying the effectiveness of several techniques should be helpful for the research community.

**Clarity**
- The overall writing is clear. The authors utilize figures well to illustrate the ideas and framework. Figure 3 clearly shows an overview of the network architecture.
- The paper gives clear descriptions in both theoretical and intuitive ways. The notations and the formulations are well-explained.

**Ablation study**
Ablation studies are comprehensive. The proposed framework consists of multiple components. The provided ablation studies are helpful for analyzing their effectiveness of them.

**Experimental results**
The presentation of the experimental results is clear. Particularly, Figure 7 provides clearly understandable results showing pre-trained representations from Q-learning enable positive transfer to novel games and lead to significant performance gain compared to return-conditioned supervised learning methods and representation learning approaches.

**Reproducibility**
Given the clear description in the main paper and the details provided in the appendix, I believe reproducing the results is possible.

## Paper weaknesses and questions

**Experiment**
The experiments could be more diverse. While the claims and the results look promising, the proposed method is only evaluated in Atari games. I believe evaluating in robot manipulation domains, locomotion domains, or navigation domains would make this work a lot stronger.

**Summary Of The Paper:**

This paper aims to scale up model capacity and improve the generalization performance across tasks with offline Q-learning methods. In contrast, prior works mainly centered around small-scale, single-task problems where broad generalization and learning general-purpose representations are not expected. To this end, the paper proposes a framework Scaled Q-learning that combines the following three choices (1) make modifications to the ResNet family - utilize group normalization instead of batch normalization in ResNets, and utilize point-wise multiplication with a learned spatial embedding when converting the output feature map of the vision backbone into a flattened vector which is to be fed into the feed-forward part of the Q-function, (2) leverage a distributional representation of return values and a cross-entropy TD loss for training, and (3) use feature normalization to stabilize training. The experiments show the proposed framework outperforms the baselines (decision transformers). Ablation studies suggest that (1) C51 leads to much better performance for both ResNet 50 and ResNet 101 models in terms of mean squared error and (2) adding feature normalization significantly improves performance for all the models. I am leaning toward accepting this paper because it studies a promising research direction (i.e. improving offline Q-learning) and proposes a reasonable framework with supporting experimental results. My only concern is that the experiments can be improved by adding results from different domains such as robot manipulation, locomotion, or navigation.


**Summary Of The Review:**

I am leaning toward accepting this paper because it studies a promising research direction (i.e. improving offline Q-learning) and proposes a reasonable framework with supporting experimental results. My only concern is that the experiments can be improved by adding results from different domains such as robot manipulation, locomotion, or navigation.

---

> ### Author Response · Authors · 2022-11-13
> **Author Response**
>
> Thank you for your detailed feedback and for a positive assessment of our work.
>
> Our long-term vision is to enable offline RL methods to successfully train large models on large datasets in a variety of problem domains. We agree with you that the natural next step would be to evaluate and extend our method to other domains such as robotic manipulation, navigation, locomotion.
>
> For our current study, we had to balance breadth vs depth with our compute budget, and we decided to focus on Atari for a few reasons. Multi-game Atari is an established open-source, challenging test bed for Q-learning (also noted by **Reviewer k6mY**), where Lee et al. 2022 showed that naively applying offline Q-learning performs poorly. This raises the question of whether Q-learning can scale and generalize from large amounts of diverse data at all.  On the other hand, most widely used benchmarks for other domains tend to be smaller. We do believe that developing large-scale tasks of this sort and scaling offline RL to these domains is the next important step, and we have added this as an avenue for future work in our discussion (Section 6). We believe that our results provide a good starting point for scaling offline Q-learning on other problem domains.
>
> Finally, we would like to note the significance of our results, even though we study only the multi-game Atari domain. For the first time, our results show that scaled Q-learning can outperform prior supervised learning methods at such large scale ( **2x** performance improvement).  As a testament to the difficulty of this problem, we note that prior works that run online RL (Espeholt et al. 2018, Teh et al. 2017) often need to employ techniques such as pre-training, distillation, or supervised auxiliary losses to attain good performance on multi-game Atari. We show that a method based entirely on offline Q-learning can scale to such settings without requiring auxiliary losses or distillation procedures. Overall, our experimental results indicate that scaled Q-learning provides a solid starting point to study future directions of scaling Q-learning.

---

> > ### Comment · Reviewer_oQ8T · 2022-11-25
> > **Re: Author Response**
> >
> > I appreciate the author's rebuttal. After carefully reading other reviewers' comments, I have decided to keep my original score.

---

> ### Author Response · Authors · 2022-11-16
> **Follow-up**
>
> Dear Reviewer oQ8T,
>
> Thank you for your detailed feedback! As the rebuttal period will close in 2 days, we were wondering if you had any other questions that you would want us to answer. We would be happy to discuss!
>
> Thanks!!

---

### Official Review · Reviewer_PcZp · 2022-10-25

**Confidence:** 3
**Correctness:** 4
**Technical Novelty And Significance:** 4
**Empirical Novelty And Significance:** 4
**Recommendation:** 10

**Clarity, Quality, Novelty And Reproducibility:**

The paper is well written, the experiments are high quality, the results are novel and the experiments can be reproduced thanks to the thorough descriptions in the appendix.  Due to the lack of error bars it is unclear, however, that the results of such a reproduction would be the same (though the size of the improvements of their proposed method are large enough that it is unlikely variance would substantially change the conclusions).

**Strength And Weaknesses:**

Strengths:
- The obtained results are very impressive, from the effectiveness of the methods to enable strong scaling trends, to the ability to dramatically surpass the scores of the training trajectories, to the strong online and offline fine-tuning results.
- The proposed modifications were well motivated and the ablations illustrate the contribution of each change.
- Well written, with formatting that makes it easy to follow the key contributions and points.

Weaknesses:
- None of the results except Figure 8 have error bars despite the fact that the error bars in Figure 8 indicate there may be substantial variance.
- As far as I can tell, the MT Impala-DQN* setting is not described in/around Figure 5.


**Summary Of The Paper:**

This paper targets harnessing the potential of offline RL to allow high-capacity models to be trained on large datasets from a range of tasks, enabling generalization.  They propose modifications to standard offline Q-learning setups that enable performance to scale with model capacity and then demonstrate the effectiveness of these modifications through thorough experimentation.  They successfully train a single model on 40 Atari games and demonstrate that the model is able to outperform humans on most of the games, dramatically outperform other offline methods when trained on suboptimal data, and also provide a strong initialization for fine-tuning on harder variants of the individual games.


**Summary Of The Review:**

This paper provides a major contribution towards being able to use large datasets and high capacity models, which have been very effective in other domains, to solve RL problems.  The proposed changes are clear and the results are very compelling, I strongly recommend acceptance.

---

> ### Author Response · Authors · 2022-11-13
> **Author Response**
>
> Thank you for your detailed feedback, and a positive assessment of our work! We are glad that you found that this paper provides a major contribution.
>
> **MT Impala DQN**: We have now updated the paper to clearly describe the MT Impala DQN baseline (**Figure 1**, **Appendix B.5**) – this baseline was taken directly from Lee et al. 2022 (Multi-game decision transformers) and corresponds to training a multi-task DQN online on all the 40 games, with an IMPALA architecture, following Espeholt et al. 2018.
>
> **Error bars** – To ensure that our large-scale results exhibit lower variance, we followed the protocol from Lee et al. 2022, and utilized Atari environments _without_ sticky actions for evaluating our trained agents. This significantly reduces variance, and enables us to make concrete evaluations. For Figure 8, we followed the protocol from Farebrother et al. 2021, which requires us to use sticky actions and this increases variance in Atari evaluation. Hence, we reported error bars for Figure 8 specifically.
>
> We hope that our answers clarify your concerns.  Thank you for your time and feedback!

---

### Official Review · Reviewer_YtKE · 2022-10-27

**Confidence:** 5
**Correctness:** 4
**Technical Novelty And Significance:** 3
**Empirical Novelty And Significance:** 3
**Recommendation:** 8

**Clarity, Quality, Novelty And Reproducibility:**

Very clear. High quality. Very novel. Hard to reproduce, but for large-scale studies like this, we can't ask for too much.

**Strength And Weaknesses:**

Strength
- The paper is very well-written.
- Design choices are well-explained and motivated.
- Experiments/ablations are abundant, and experimental results are convincing.
- The paper offers very convincingly results that offline RL algorithms can train very large models on a massive amount of data. This paves the way for a large unified decision-making foundation model (so to speak).

Weakness
- As thorough as this paper is, I find it relatively disappointing that despite many different offline RL algorithms having been proposed over the last 2-3 years, the authors only used CQL. This opens up several interesting empirical questions:
  1. The paper makes it seem that DR3 (feature normalization), categorical representation of return values, and Q-function architecture matter a great deal. Does the offline learning algorithm matter at all? Would TD3 + BC also scale? Would BCQ scale? It's definitely difficult to evaluate every single offline RL algorithm, but it would be great if at least one other offline RL algorithm is chosen to compare with CQL.
  2. With a huge amount of data (400M transitions and 2B transitions), one might wonder if pessimism is no longer needed since all possible (s, a) might have already been covered -- is the pessimistic penalty really necessary? An ablation study can train with TD3 directly or any off-policy Q-learning algorithm.
- "...making too many TD updates to the Q-function in offline deep RL is known to sometimes lead to performance degradation and unlearning" [1]. On such a large dataset with 40 games, is there an optimal stopping point for training? Is there performance degradation if you train longer? Or do you think because of DR3, performance degradation won't happen anymore?

[1] DR3: VALUE-BASED DEEP REINFORCEMENT LEARNING REQUIRES EXPLICIT REGULARIZATION

**Summary Of The Paper:**

This paper looks into whether carefully making design choices for Offline RL training objectives can have similar power law scaling to supervised learning.

**Summary Of The Review:**

I think this paper's quality far surpasses the current score I'm giving (which is a 5). I'm giving this score because I think the authors (with access to the amount of computing power and resource) should address some of the concerns I raised. A simple rebuttal would be, "investigating different kinds of offline RL training algorithms is out of the scope of this paper" -- this I fully understand, but the authors have a great experimental setup, and a massive amount of data, which I doubt many other people have access -- some preliminary investigations on my questions would benefit the Offline RL community tremendously.

---

> ### Author Response · Authors · 2022-11-13
> **Author Response: Added requested experimental studies**
>
> We thank the reviewer for their detailed comments! We are glad that they found our paper clear, novel and of high quality. To address their concerns, we have updated the paper to add new experimental results (**Appendix A**) studying the choice of the offline RL algorithm (BCQ), and the necessity of conservatism. To further facilitate future work, we have now open-sourced our code in the supplementary. Our experiments use the already public [DQN-replay](https://offline-rl.github.io/) dataset.
>
> **We hope that most of the reviewer’s concerns have been addressed and, if so, they would reconsider their assessment. We’d be happy to engage in further discussions.**
>
> ___
>
> > **Does the choice of offline RL algorithm matter? It would be great if at least one other offline RL algorithm is chosen to compare with CQL.**
>
> This is a great question! While we had access to a large compute budget, it was still limited, so we had to choose between depth and breadth. As our goal was to demonstrate that offline Q-learning *can* scale and generalize, we focused on a single algorithm allowing us to conduct a more systematic analysis. We chose CQL because it is easy-to-implement, performs well, and importantly, only requires two networks (Q and target Q), like DQN / C51, substantially reducing the memory burden compared to other offline RL methods which require training additional policy or value networks (e.g., a behavior policy).
>
> As requested, we have started scaling experiments for discrete BCQ in conjunction with our other improvements (C51, feature normalization, our ResNet architecture). We have completed this study for the 6-game setting and these results are presented in detail in **Appendix A.2** in **Figure A.3** and copied below.
>
> For the 6-game setting with discrete BCQ, the results are shown in the **[anonymous figure](https://i.imgur.com/u2dh64Z.png)**. We find that **scaled QL with discrete-BCQ can also scale**, and the performance improves from ResNet 34 to ResNet 50. Unfortunately, we only had temporary access to large-scale compute, and with our current compute budget, the complete 40-game discrete-BCQ study will take around 2 months, and we will add it to the final version.
>
> **Other algorithms.** We cannot directly apply TD3+BC because TD3+BC is designed for continuous-control and Atari is discrete-control. In the discrete-control setting, CQL combines Q-learning with a BC loss on the softmax policy induced by the Q-function.
>
> ___
>
> > **Is pessimism no longer needed since all possible (s, a) might have already been covered?**
>
> Thank you for this suggestion. To evaluate the impact of pessimism, we ran an experiment comparing scaled QL with and without CQL loss in the 6 game setting, which contains the same state-action pairs for each of the 6 games as in our 40 game study. We have added these results in **Table A.2** in **Appendix A.4**, and present them below.
>
> | **Games**         | **Without CQL**   | **With  CQL**    |
> | ------------- | ------ | ------ |
> | Asterix       | 38000  | 35200  |
> | Breakout      | 322.6  | 410.6  |
> | Pong          | 12.6   | 19.8  |
> | Qbert         | 13800  | 15500  |
> | Seaquest      | 1378   | 3694   |
> | SpaceInvaders | 1675   | 3819   |
> | **Human-normalized IQM**           | 188.3% | 223.4% |
>
> Observe that the addition of the CQL loss improves performance. Although scaled QL without CQL loss does learn (unlike divergence typically observed without pessimism on narrow datasets), using CQL still outperforms it.
>
> We have launched this study in the full 40-game setting, but have so far only observed poor performance without pessimism. These observations are preliminary, and we will run it for longer and update the final paper with the results.
>
> ___
>
> > **Is there an optimal stopping point for training? Is there performance degradation if you train longer?**
>
> This is a great question! Following Lee et al. 2022, we report performance after 10 million gradient steps, for which training took around 3 weeks of wall-clock time. We did **not** observe performance degradation within this much training. We suspect that performance might improve with further training, but we are limited by compute.
>
> Prior results in single-game Atari settings only observe performance drops significantly later as the size of the dataset increases (e.g., see Figure 2, Kumar et al. 2021, where the drop with 4x more data appears only after 6 million gradient steps). Our 40-game dataset is 200 times bigger than the “4x” data in Kumar et al. 2021, which suggests that if at all, we might observe a drop after training for several months longer.
> ___

---

> > ### Comment · Reviewer_YtKE · 2022-11-13
> > **Satisfied**
> >
> > I'm satisfied with this additional investigation and I applaud the authors' willingness to engage with the reviewer community! I encourage the authors to include these results and discussions in the appendix of the paper, as I think many will find them interesting.
> >
> > I'm increasing my score to reflect my current level of satisfaction with this work.

---

> > > ### Author Response · Authors · 2022-11-16
> > > **Thank You!**
> > >
> > > Thank you very much for engaging with us and for increasing your score! We really appreciate it!
> > >
> > > We will make sure to add all of these studies and discussion in the appendix of the paper.

---

### Decision · Program_Chairs · 2023-01-20

**Decision:**

Accept: notable-top-5%

**Justification For Why Not Higher Score:**

N/A

**Justification For Why Not Lower Score:**

Since all the reviewers agree with the novelty and solid intuition behind the proposed method and in-depth experimental evaluation, I would like to choose Accept with Oral.

**Metareview: Summary, Strengths And Weaknesses:**

This paper aims to scale up model capacity and improve the generalization performance with offline reinforcement learning. In particular, the proposed design choices include making modifications to ResNets, leveraging cross-entropy based distributional backups and utilizing feature normalizations. Experimental results demonstrate that the effectiveness of the methods to enable strong scaling trends and the ability to dramatically surpass the scores of the training trajectories.

Overall, the paper is well written. The design choices are well explained and motivated. The experimental results and ablation studies are solid. All the reviewers agree with the novelty and solid intuition behind the proposed method and in-depth experimental evaluation.


**Note From Pc:**

if the above contains the word "oral" or "spotlight" please see: "oral" presentation means -> notable-top-5% and "spotlight" means -> notable-top-25%. As stated in our emails, we are disassociating presentation type from AC recommendations